# Facile Fabrication of Multi-Hydrogen Bond Self-Assembly Poly(MAAc-co-MAAm) Hydrogel Modified PVDF Ultrafiltration Membrane to Enhance Anti-Fouling Property

**DOI:** 10.3390/membranes11100761

**Published:** 2021-09-30

**Authors:** Weigui Fu, Guoxia Li, Gaowei Zhai, Yunji Xie, Meixiu Sun, Patrick Théato, Yiping Zhao, Li Chen

**Affiliations:** 1State Key Laboratory of Separation Membranes and Membrane Processes, School of Materials Science and Engineering, Tiangong University, Tianjin 300387, China; libra951001@163.com (G.L.); zhui02648484@163.com (G.Z.); yipingzhao@tjpu.edu.cn (Y.Z.); chenlis@tjpu.edu.cn (L.C.); 2Institute for Chemical Technology and Polymer Chemistry (ITCP), Karlsruhe Institute of Technology (KIT), Engesserstraße 18, 76131 Karlsruhe, Germany; yunjixie512@gmail.com (Y.X.); Patrick.theato@kit.edu (P.T.); 3Institue of biomedical Engineering, Chinese Academy of Medical Science & Peking Union Medical College, Tianjin 300192, China; meixiu_sun@126.com; 4Soft Matter Synthesis Laboratory, Institute for Biological Interfaces 3 (IBG-3), Karlsruhe Institute of Technology (KIT), Herrmann-von-Helmholtz-Platz 1, 76344 Eggenstein-Leopoldshafen, Germany

**Keywords:** surface modification, UV grafting polymerization, antifouling, poly(MAAc-co-MAAm) hydrogel, supramolecular assembly, organic foulants

## Abstract

In this work, a facile preparation method was proposed to reduce natural organics fouling of hydrophobic membrane via UV grafting polymerization with methacrylic acid (MAAc) and methyl acrylamide (MAAm) as hydrophilic monomers, followed by multihydrogen bond self-assembly. The resulting poly(vinylidene fluoride)-membranes were characterized with respect to monomer ratio, chemical structure and morphology, surface potential, and water contact angle, as well as water flux and organic foulants ultrafiltration property. The results indicated that the optimal membrane modified with a poly(MAAc-co-MAAm) polymer gel layer derived from a 1:1 monomer ratio exhibited superior hydrophilicity and excellent gel layer stability, even after ultrasonic treatment or soaking in acid or alkaline aqueous solution. The initial water contact angle of modified membranes was only 36.6° ± 2.9, and dropped to 0° within 13 s. Moreover, flux recovery rates (FRR) of modified membranes tested by bovine serum albumin (BSA), humic acid (HA), and sodium alginate (SA) solution, respectively, were all above 90% after one-cycle filtration (2 h), significantly higher than that of the pure membrane (70–76%). The total fouling rates (*R*_t_) of the pure membrane for three foulants were as high as 47.8–56.2%, while the *R*_t_ values for modified membranes were less than 30.8%. Where *R*_t_ of BSA dynamic filtration was merely 10.7%. The membrane designed through grafting a thin-layer hydrophilic hydrogel possessed a robust antifouling property and stability, which offers new insights for applications in pure water treatment or protein purification.

## 1. Introduction

Membrane separation technology shows great potential in sewage treatment for its easy operation, energy saving, and environmental friendliness [1,2,3]. Poly(vinylidene fluoride) (PVDF) ultrafiltration (UF) membranes were widely used in the chemical, biomedicine, and food industries, as well as for drinking water or wastewater treatment, thanks to its good mechanical properties, excellent chemical resistance, and thermal stability [4,5,6]. However, organic foulants, like bovine serum albumin (BSA), humic acid (HA), and sodium alginate (SA) [7,8,9,10,11,12,13,14], easily adsorb onto the hydrophobic membrane surface or block into internal pores during the filtering process, thus reducing water flux and sacrifice membrane lifetime [15,16]. Forming hydration layers to enhance the hydrophilicity of the membrane surface is an effective way to reduce organics fouling [17,18]. 

Typical hydrophilic modification methods include copolymerization, blending, and surface modification by coating or grafting [19,20,21,22,23,24]. Chen et al. prepared a novel amphiphilic copolymer PVDF-g-PMABS by introducing a hydrophilic additive into the PVDF matrix to fabricate porous blending membranes [19]. It was found that the membrane exhibited a remarkable FRR of 98.6%, whereas the total fouling rate (Rt) is about 30%. However, coatings are easy to detach from membrane surfaces after long-term filtration, and the anti-protein ability of the modified membranes will dwindle drastically. Fu et al. blended sulfonated PTFS into PVDF membrane to improve the water flux and fouling resistant property. In general, the hydrophilc polymers detatch easily from the membrane materials during the filtration process [20]. In recent years, surface grafting techniques (plasma treatment, high-energy or UV-photo grafting, etc.) have been utilized effectively to improve the stability of the top layer [18,23]. Deng et al. grafted acrylic acid (AAc) or methacrylic acid (MAAc) onto PVDF powders by pre-irradiation treatment [18]. The contact angle of the modified membranes decreased from 67° to 34°, and the flux recovery rate (FRR) has improved to 90%. However, high-energy or plasma treatment is not suitable for large scale application or mass production. In contrast, a UV-photo initiated hydrophilic hydrogel layer is easy to synthesize, low cost, and without the need for metal catalysts. Besides, the three-dimension network structure grafted onto the membrane surface tends to form a stable hydration layer and is beneficial to reduce organic-fouling [21,22]. Zhang et al. grafted a novel zwitterion polyampholyte hydrogel onto the surface of polyethersulfone (PES) UF membrane [21]. After a long-term cyclic filtration experiment (2 h) of BSA, the modified membrane revealed an FRR of 85%. However, the addition of chemical crosslinking agents may introduce toxicity and increase brittleness of the top gel layers [23]. Liu et al. reported the performances of zwitterionic monomer ([2-(methacryloyloxy)ethyl] dimethyl-(3-sulfopropyl)) grafted PVDF membrane surfaces, which showed great advantages to treat waste water containing electrolyte, FRR was up to 99.3% [22]. It was noteworthy that the structural stability of zwitterionic layers was easily influenced by pH, salt concentration and so on, thus affecting the water flux and anti-fouling property of the modified membranes [24].

In our previous work, a highly hydrophilic poly(*N*-acryloylglycine) (PNAGA) hydrogel was decorated onto PVDF UF membranes by UV initiated grafting without using any chemical crosslinkers [23]. The FRR of the membrane maintained at 99% after four-cycle filtration test, while the *R*_t_ was 30% at pH 7.0. Besides, it showed long superior stability after ultrasonic 30 min and 7 days pure water filtration tests, which indicated that the gel layer grafted onto membrane surfaces through covalent bonds and its cross-linking structure based on supramolecular self-assembly played key points. However, self-polymerization occurred easily in the process of NAGA monomer synthesis. In addition, the PNAGA hydrogel layer showed relatively high swelling degree for the good hydrophilicity of the double-amide on the side chains, thus decreasing the pore size and porosity, and reducing water flux dramatically. As the optimal membrane (M2) with grafting degree 1.29 mg/cm^2^ for an example, its pure water flux decreased by 83.2% compared with that of pristine membrane. 

Hydrophilic monomers like acrylic acid (AAc) methacrylic acid (MAAc) and acrylamide (AAm) were applied to synthesize hydrogels or decorate membrane surfaces [25,26,27,28]. Wang et al. found that the stability and mechanical properties of the physical cross-linked hydrogels like poly(MAAc-*co*-AAm) and poly(AAc-*co*-MAAm) based on mutihydrogen bond are weak. However, poly(MAAc-*co*-MAAm) hydrogel formed by hydrogen bonding of inter- or intramolecular sites between oxygen atom of carboxy groups (C=O) on MAAc construction unit and proton of amino-fragments (N-H) on MAAm construction unit, without adding chemical crosslinkers, exhibited high stability under acid or weak alkaline conditions (1.6 ≤ pH ≤ 9.6), which is expected to minimize organics fouling [28]. 

In this work, methacrylic acid (MAAc) and methacrylamide (MAAm) were utilized as grafting monomers to form an ultrathin hydrophilic hydrogel layer based on mutihydrogen bond assembling on the surface of commercial flat-sheet PVDF membranes by surface-initiated UV graft polymerization for enhancing organic-fouling resistance and stability. The physicochemical properties of the membrane were characterized by a series of experiments: the stability of the hydrogel layer was examined via ultrasonic treatment or acid-alkali solution immersion; and the antifouling properties were investigated by static adsorption of BSA and three-long-period filtration tests for three organics, BSA, HA, and SA, respectively. In this study, an antifouling PVDF UF membrane with favorable hydrophilicity and stability was designed and prepared via a facile as well as green process, which was expected to provide new thoughts for designing novel fouling resistant membranes used in the fields of food processing, protein concentration, and drinking water treatment.

## 2. Experimental Section

### 2.1. Materials

PVDF membranes (mean pore size is 100 nm) were obtained from Haiyan Co. Ltd. (Zhejiang, China). Methacrylamide (MAAm) and benzophenone (BP) were purchased from Mackkin Biochemical Co. Ltd. (Shanghai, China). Methanol and disodium hydrogen phosphate dodecahydrate (Na_2_HPO_4_·12H_2_O) were supplied by Fengchuan Chemical Reagent Technologies Co. Ltd. (Tianjin, China). Methacrylic acid (MAAc) and humic acid (HA) were obtained from Tianjin Guangfu Fine Chemical Research Institute (Tianjin Guangfu Fine Chemical Research Institute, China). Ammonium ferrous sulfate ((NH_4_)_2_Fe(SO_4_)_2_·6H_2_O) was purchased from Tianjin Beifang Tianyi Chemical Reagent Factory (Tianjin, China). Bovine serum albumin (BSA, *M*_w_ = 6.7 kDa) was supplied by Bomei Biotechnology Co. Ltd. (Hefei, China). Sodium alginate (SA) was obtained from Sinopharm Chemical Reagent Co. Ltd. (Shanghai, China). Sodium chloride (NaCl), potassium chloride (KCl), potassium dihydrogen phosphate (KH_2_PO_4_) hydrochloric acid (HCl), and sodium hydroxide (NaOH) were obtained from Tianjin Kermel Chemical Reagent Co. Ltd. (Tianjin, China). 

### 2.2. Membrane Modification

Grafting modification was made by UV-light irradiation method [24]. As shown in Figure 1, commercial PVDF membranes were rinsed with methanol, and then were immersed in a benzophenone (BP) solution (0.4 mol/L) for 1.5 h, with methanol as solvent. Next, the membranes were taken out and dried in air at room temperature about 1 h. Secondly, the aqueous solution of different monomer concentrations bubbled by nitrogen for 30 min. And then the prepared membranes were irradiated immersed in the solution with UV light (365 nm, 8 W) for 30 min at room temperature. Finally, modified membranes were removed and cleaned with methanol and water mixed solution to remove the unreacted monomers, self-polymerization polymers, and BP. Some milk white flocculates could be seen on the dry modified membarens. The hydrophilicity of the membrane surface was optimized by grafting degree and monomer concentration. Firstly, to obtain an appropriate grafting degree, two monomers with the same molar ratio but different feeding amounts were used to modify membrane surfaces, and then evaluated by the water contact angle test (Appendix A in Supporting Information, SI). Next, the membranes were decorated with polymers from different molar ratios of monomers (MAAc:MAAm was 0:0, 3:1, 2:2 and 1:3) but with similar grafting degree were prepared, named as M0, M3-1, M2-2, and M1-3, respectively. Grafting degree (mg·cm^−2^) was measured by gravimetric method and follow Equation (1) listed below [21]:(1)$$GD=\frac{M_{2}-M_{1}}{S}$$
where *M*_1_ (mg) is the weight of original membrane, *M*_2_ (mg) is the weight of modified membrane, *S* (cm^2^) is the area of membrane (19.6 cm^2^).

The equilibrium swelling degree of the hydrogel layer on different samples was measured by the weight ratio before and after water adsorption for the dry hydrogel layer grafted on the PVDF films (without pores in the cross-section); further descriptions are shown in the SI.

### 2.3. Membrane Characterization 

Attenuated total reflection Fourier transform infrared spectrum (ATR-FTIR, Nicolet iS50, Thermo Fisher Scientific, Waltham, MA, USA) was used to characterize chemical composition of the membrane surface in the range of 400–4000 cm^−1^. X-ray photoelectron spectroscopy (XPS, Thermo Fisher K-alpha, Waltham, MA, USA) was done to analyze the elemental composition. 

Scanning electron microscopy (SEM, Hitachi S4800, HITACHI, Tokyo, Japan) was utilized to describe the surface and cross section morphology of membranes. Prior to the test, the membranes ought to be dry under vacuum at 25 °C, and then were sputter-coated with gold.

Roughness of the dry and wet membrane surfaces was tested three times repeatedly at different places (scanning area was 5 μm × 5 μm) by atomic force microscopy (AFM, Angilent 5500, Angilent, Santa Clara, CA, USA) with a tapping mode at room temperature. 

The surface charge of membranes was calculated based on outer surface streaming potential (Zetaszier, SurPASS 3, Austria) within the pH = 3~9 (adjusted with 0.1 M HCl and 0.1 M NaOH) in a 1 mM KCl solution at a temperature of 25 ± 1 °C (Described in SI). Two samples were arranged parallel to each other and were spaced apart 100 μm. 

Dynamic water contact angle (WCA) of the samples was measured by the drop shape analysis system (Krüss DSA100, Krüss Scientific, Hamburg, Germany). The volume of distilled water placed on the top surface of membranes was 2 μL, and the digital image was recorded by the camera of the test instrument. Measurements were taken from three randomly selected spots and the average WCA was reported.

To characterize the cross-linking degree of different modified membranes. The swelling degree (*S*) of the hydrogel layer on the samples was measured by the weight of the corresponding hydrogel layer on the PVDF films without pores in cross-section before and after adsorbing water to equilibrium. The swelling degree of the hydrogel layer was calculated based on the Equation (2) [22]:(2)$$S=\left(\frac{W_{wet}-W_{dry}}{W_{dry}}\right)\times 100\%$$
where *W*_wet_ (g) is the weight of wet film after soaking in deionized water for 24 h, *W*_dry_ (g) is the weight of dry film. 

The porosity (*ε*) and mean pore size (*r*_m_) of these samples were analyzed by the dry-wet membrane method [29]. Firstly, the weight of the drying membrane was recorded. and then the membrane was put into the deionized water until the adsorption equilibrium. Next, removed the membrane and wiped the adsorbed water on the surface. After that, recorded the weight of the wet membrane was weighed. Here, *ε* is calculated by Equation (3):(3)$$\varepsilon \;(\%)=\frac{W_{w}-W_{d}}{\rho Al}\times 100$$
where *W*_w_ (g) is the weight of wet membrane, *W*_d_ (g) is the weight of dried membrane, *ρ* (0.998 g·cm^−3^) is water density, *A* (cm^−2^) represents the area of membrane surface, *l* (cm) is the thickness of wet membrane.

The mean pore size was calculated as Equation (4):(4)$$r_{m}=\sqrt{\frac{(2.9-1.75\varepsilon )\times 8\eta lQ}{\varepsilon A\Delta P}}$$
where *η* (8.9 × 10^−4^ Pa·s) is water viscosity, *Q* (L/s) is the volume of permeate water, Δ*P* (0.1 MPa) is transmembrane pressure.

### 2.4. Evaluation of Gel Layer Stability

Ultrasonic and immersing in an acid or alkali solution tests were used to characterize the stability of hydrogel layers. The membrane samples were placed in an ultrasonic cleaner (KQ5200E, China) for 30 min, and immersed in a HCl solution (pH = 2) and a NaOH solution (pH = 9) for 30 min, respectively. Afterwards, the membranes were taken out and dried. Finally, membrane weights of before and after treatment were compared.

### 2.5. Fouling Properties

#### 2.5.1. Static Adsorption of Protein 

To understand the antifouling property of postmodified membranes with respect to the original membrane, static adsorption experiments were conducted with BSA (concentration of 0.5 g/L) [30]. Round membranes with a radius of 2.5 cm were immersed in 50 mL of BSA solution and shaken at 25 °C for 14 h to achieve adsorption equilibrium. The concentration (μg/cm^2^) of BSA solution before and after membrane immersion were measured with a UV-1800 at a wavelength of 278 nm, and the protein adsorption amount on the membrane surface was calculated using Equation (5):(5)$$Q=\frac{\left(C_{0}-C_{1}\right)\times V}{S}$$
where *Q* (μg/cm^2^) is the amount of adsorbed BSA, *V* (L) is the volume of protein solution, *S* (cm^2^) is the effective adsorption area of membranes, and *C*_0_ (μg/L) and *C*_1_ (μg/L) are the concentration of BSA solution before and after adsorption, respectively. 

#### 2.5.2. Dynamic Fouling Experiments

Three-cycle filtration test was used to characterize the antifouling ability of the membrane surface by a cross-flow device, and solution of BSA, HA and SA was used as foulants. BSA solution was prepared from phosphate buffer solution (PBS, pH = 7.4). HA solution was made by dissolving HA powder into DI-water (pH = 11), then adjusted the solution pH value to 7 after filtering the undissolved portion. SA solution was prepared by dissolving 100 ppm sodium alginate into 10 mM NaCl solution. Firstly, the membrane sample with an effective area of 13.85 cm^2^ was compacted for 0.5 h at 0.2 MPa. The pressure was then changed to 0.1 MPa to obtain the pure water flux (*J*_w0_), and a value was recorded every 5 min for a test time of 30 min. Then, the pure water was replaced by feed solution with foulants for 2 h, and the flux of foulant solutions (*J*_P_) was recorded. A simple hydraulic flushing was performed on the fouled membrane, and the pure water flux (*J*_w1_) of the membrane was measured again. The next two cycles were repeated as above, and the corresponding pure water flux is named as *J*_w2_ and *J*_w3_, respectively. 

The flux (*J*, L·m^−2^·h^−1^) was calculated using Equation (6) [31]:(6)$$J=\frac{V}{A\Delta t}$$
where *V* represents the volume of liquid passed through the membrane (L), *A* is effective area of membrane (m^2^), and △*t* represents the permeated time (h).

Besides, flux recovery rate (FRR) represents the reusability of membrane after filtration, the total fouling rate (*R*_t_) is the degree of flux reduction during the filtration, the reversible fouling rate (*R*_r_) and the irreversible fouling rate (*R*_ir_) are the degree of flux reduction renewable and unrecoverable, respectively. All the above parameters should be considered to characterize the fouling resistant performances, and can be calculated by the following Equations (7)–(10), respectively [32]:(7)$$FRR_{n+1}=\left(\frac{J_{Wn+1}}{J_{Wn}}\right)\times 100\%\;(n=0,1,2)$$
(8)$$R_{t}=\left(\frac{J_{W0}-J_{P}}{J_{W0}}\right)\times 100\%$$
(9)$$R_{ir}=\left(\frac{J_{W0}-J_{W1}}{J_{W0}}\right)\times 100\%$$
(10)$$R_{r}=\left(\frac{J_{W1}-J_{P}}{J_{W0}}\right)\times 100\%$$

## 3. Results and Discussion

### 3.1. Surface Characterization of Membranes

ATR-FTIR spectra were performed on the surface of pristine and modified membranes to validate the physicochemical properties of the membrane (Figure 2). The characteristic peak at 1168 cm^−1^ was ascribed to C-F stretching modes of PVDF [33], the peak strength of the modified membranes decreased compared with that of the pure membrane (M0). In comparison with that of the pure membrane, there were three new peaks in the modified membranes, 1700 cm^−1^, 1650 cm^−1,^ and 1600 cm^−1^, corresponding to the stretching of the C=O bond in MAAc, the stretching vibration of C=O, and N-H bonds in MAAm, respectively [13]. The peak of O-H vibration absorption of hydrogen bonds between polymers was observed around 3398 cm^−1^. Notably, the intensity of peaks at 1650 cm^−1^ and 1600 cm^−1^ increased, while the peak at 1700 cm^−1^ decreased.

As shown in Figure 3, XPS spectra quantified the compositions of the upper-surface for pure PVDF membrane (M0) and the modified membranes (M3-1, M2-2, and M1-3). In Figure 3A, strong emissions at 532.0 eV (O 1s) and 400.0 eV (N 1s) confirmed the existence of PMAAc and PMAAm chains on the modified membrane surface. Figure 3B exhibited the C core-level 1 s spectra of membranes. Compared with that of M0, modified membranes displayed new binding energies at 288.6 eV ascribed to O-C=O of PMAAc and 286.2 eV assigned to C-N of PMAAm [34]. The relative contents of elements are summarized in Table 1, with the increased of MAAc mass ratio, the percentage of N element increased, and the corresponding percentage of O element decreased. The results of ATR-FTIR and XPS prove that PMAAc and PMAAm were grafted on the membrane surface successfully.

### 3.2. Morphology of Membranes

Membrane surface and cross section SEM images of pure membrane (M0) and modified membranes (M3-1, M2-2, and M1-3) are shown in Figure 4. M0 possessed a loose spongy structure, and the spongy layer of the modified membranes became compact. The complete P(MAAc-co-MAAm) gel layer on the modified membarne surfaces can not be observed obviously, but the pore size decreased; that is to say, the gel layer surrounded the fibers on the membrane surface or was partially grafted in the pore walls. According to the surface images (×10,000) of the membranes, the thickness of the gel layer was less than 100 nm. It is consistent with the calculation results of pore size (Appendix A). Notably, the ultra-thin gel layer on the top of the PVDF substrates does not cause dramatic flux reduction.

The roughness of membrane surfaces is one of the key factors affecting the antifouling ability. AFM images of dry and wet membrane surfaces are shown in Figure 5. The mean roughness (*R*a) of the modified membranes (M3-1, M2-2, M1-3) decreased compared with that of the dry original membrane (M0, 230 nm), attributed to the coverage of gel layers on the substrates. Under the same grafting degree, *R*a of the modified membranes (M3-1, M2-2, M1-3) increased from 170 to 223 nm with increasing PMAAm content, which may be due to the lower reactivity of PMAAm. As can be seen from the *R*a of PVDF-g-PMAAc membrane and PVDF-*g*-PMAAm membrane (Appendix A), the latter showed a higher surface roughness. Therefore, M1-3 with a relatively high roughness (*R*a= 223 nm) probably had a lower grafting density and relatively longer molecular chains, which may cause long grafted chains agglomeration. Moreover, *R*a of wet pristine membrane (M0’, 225 nm) changed little compared to that of the hydrophobic dry sample (M0, 230 nm), for the wet pristine PVDF (M0’) flat sheet membrane was resistant to swelling, whereas the modified membranes after absorbing water (M3-1’, M2-2’, M1-3’) possessed lower roughness (150-179 nm) than that of the corresponding dry membranes. That may be because the membrane surface became smoother when the wet top layer reaches swelling equilibrium. Normally, membranes with higher roughness tend to adsorb more foulants [35]. Effects of membrane roughness on swelling degree and antifouling performance were subsequently verified. 

### 3.3. Zeta Potential of Membranes

The surface charge has a significant influence on the antifouling ability of membranes. Generally, membrane surfaces with negative charge have better resistance to negatively charged foulants [8,15]. Figure 6 shows the change of zeta potential of the original membrane (M0) and the modified membranes (M1-2, M2-2, and M3-1) as a function of pH value. All the membranes possessed negative zeta potential at pH = 7, and the absolute value of M0 was maximum due to the larger surface electronegativity of F atoms [25]. The zeta potential of modified membranes covered by hydrogel layers decreased as the monomer content of MAAm increases. It is attributed to a reduction of negatively charged carboxyl groups (COOH), as well as a large increase in amino groups (NH) with positive charges on the membrane surface. 

### 3.4. Surface Hydrophilicity and Swelling Degree

The better the hydrophilicity of membrane surfaces, the weaker the adhesion between membrane surfaces and foulants. Time-dependent water contact angles (WCAs) are shown in Figure 7a. The WCA of the pure membrane maintained about 116.7° over the range from 0 to 30 s, while the initial WCA of the modified membranes was much lower and decreased dramatically with increasing time. Notably, M2-2 exhibited better hydrophilicity with the lowest initial contact angle (36.6°), and it took 13 s to reduce WCA to 0°. Generally, when the WCA of the same material was within 90°, it decreased with the increasing of the roughness, which demonstrates that the material surface is easily moistened. Regarding for the three modified membranes, as MAAm content increased, the roughness of membrane surface showed an upward trend (in Figure 5). For instance, M2-2 with a larger surface roughness displayed better hydrophilicity than M3-1 (73.8°). However, the water droplet penetration time of M2-2 was slightly shorter than that of M1-3 bearing the largest roughness, which indicates a better hydrophilicity. It is probably due to the low reaction activity of MAAm, further the small feeding amount of MAAc resulting in a low-grafting density and poor distribution uniformity of the functional layer on the M1-3 substrate.

As shown in Figure 7b, the swelling degrees of the hydrogel layers grafted on the surfaces of PVDF films (without finger-like and sponge-like pores) made by solution casting method increased with the increasing of MAAm monomer content, ranging from 6.4% (F3-1) to 10.4% (F1-3), those were much lower than that of our previous PNAGA function layer (40–65%) [23]. The result indicated that ultrathin hydrogel layers with low water content and excellent hydrophilicity were constructed, thus it helps to reduce flux losses.

### 3.5. Stability of the Grafted Functional Layer

To characterize the stability of the membranes, ultrasonic treatment and exposure to acid (pH = 2) or alkali solution (pH = 9) tests were performed, respectively. The weight of the membranes before and after ultrasound or soaking treatment (Appendix A) remained almost the same. Besides, according to the SEM images of the membrane surface, no obvious change on the surface morphology (Appendix A) was observed. Appendix A shows that there was only a minor change of water flux after all the treatments. The above results demonstrate that the excellent stability of physical crosslinked hydrogel layer grafted onto the membrane surface by UV graft polymerization, which is difficult to be destroyed by ultrasonic treatment and acid or alkali solution immersion. 

### 3.6. Antifouling Properties

Figure 8 shows the flux change and the corresponding flux recovery ratio of the pristine PVDF membrane and modified membranes in the three-cyclic filtration experiment. BSA, HA, and SA used as model foulants were applied to characterize the membrane fouling resistant property. Compared with that of the original membrane M0, the initial pure water flux of the modified membranes somewhat reduced (Figure 8A–C). The reason might be the decrease of the average pore size on the surfaces due to the coverage of the multihydrogen bond self-assembly hydrogel layers, which is in agreement with what is reported in the literature [36,37]. Among the modified membranes, M1-3 exhibited a higher flux than that of the other two modified membranes because of its largest pore size (as shown in Appendix A). Although the water flux of the membranes for feed solutions was lower than the pure water flux due to membrane fouling, all the modified membranes maintained good permeability after three cyclic filtration tests. For instance, the BSA water flux of M2-2 is 1.7 times as well as M0 during the third 2 h filtration process (Figure 8A). The BSA water flux of M1-3 is similar to the M0, and this may be because of the relatively bad hydrophilicity and relatively small pores size compared with that of the other two modified membarens. The results agree with that of the water contact angle of the membrane surface and the swelling ratio (Figure 7).

Besides, the FRR of M2-2 for BSA is 96.7% and 90.9% after the first and the third long-term filtration, respectively, while the FRR of M0 was only 42.7% after three-long-term filtration (Figure 8A’). The excellent flux recovery rate of the modified membranes illustrates that the hydrophilic hydrogel layer on the membrane surfaces has enhanced membrane hydrophilicity, thereby improving the antifouling performance of the membranes. Compared with that of M1-3, the membrane M3-1 exhibited relatively better FRR due to its higher surface potential and smaller roughness (Figure 8B’,C’).

Moreover, static adsorption tests were also applied to quantitatively characterize antifouling performance of membranes. The BSA adsorption amount of M2-2 reduced to 20.9 μg/cm^2^ from 88.0 μg/cm^2^ of that of the pristine membrane (Figure 9a). This phenomenon is consistent with the effect of roughness to the fouling resistance. The reason is that the gel layer on the membrane surface form a hydration layer after adsorbing water molecules, which effectively prevents proteins from closing to or adsorbing onto the membrane surface. Among them, M2-2 exhibited the best antiadhesive capacity due to its optimal hydrophilicity, and the static adsorption amount has reduced by 76.2%. It showed a better antifouling property compared with that of the previous work for the same BSA concentration [29,38], and those were 65.4% and 73.7%, respectively. 

The total fouling rate (*R*_t_), reversible fouling rate (*R*_r_) and irreversible fouling rate (*R*_ir_) of membranes for BSA, HA, and SA were also discussed. As shown in the Figure 9b, BSA has caused the lightest fouling on the modified membrane (M2-2), and the *R*_t_ was as low as 10.7%, and *R*_ir_ is merely 3.3%. Whereas for the PNAGA-modified membrane in our previous work, *R*_t_ and *R*_ir_ were about 30% and 1.2%, respectively [23]. The PNAGA modified membranes showed a lower *R*_ir_, probably due to its higher absolute Zeta potential (−48.4 mV) of M2 than that of M2-2 (−11.8 mV) (Figure 6), which was not conducive to the adhesion of foulants on the membrane surface. However, M2-2 prepared in this work displayed much lower *R*_t_ for all the three foulants due to its superior hydrophilicity.

Dynamic light scattering (DLS, Nano ZS90, UK) was used to measure particle size distribution of organic foulants. To make the particles disperse uniformly, supersonic treatment was applied to the BSA, HA, and SA solutions prior to the measurement, respectively, to make the particles disperse uniformly. The particle sizes of the three foulants are shown in Appendix A. Compared with that of HA and SA, BSA with the smallest particle size caused the lightest membrane fouling. Similar results showing a lighter BSA fouling than that of HA were also found by Guan et al. [13]. Both HA and SA cause relatively serious fouling, which is more likely to be trapped on the membrane surface due to their larger molecular weight and the corresponding larger particle size [39]. Moreover, the hydrophobic HA had a smaller *R*_t_ due to its smaller adhesion force with the membrane surface, and in contrast, the hydrophilic SA caused the most serious contamination [40,41].

Finally, we compared the antifouling properties of the M2-2 membranes fouled by three foulants with those reported literature, respectively. As shown in Table 2, the *R*_t_ of M2-2 membrane was merely 10.7%, and the relevant FRR was as high as 96.7% after the first long-term filtration for BSA. Regarding for HA and SA, even though *R*_t_ values were higher (24.9, 30.8), *R*_ir_ kept only 7.2% and 9.2%, respectively, as shown in Figure 7b. Overall, the M2-2 decorated with poly(MAAc-*co*-MAAm) hydrogel layer via surface-initiated UV graft polymerization exhibited a robust antifouling performance. 

## 4. Conclusions

A hydrophilic modified PVDF UF membrane with stable P(MAAc-*co*-MAAm) hydrogel layer via multihydrogen bond self-assembly was facilely prepared by UV-initiated radical graft polymerization. The ultrathin hydrogel layer grafted onto the membrane surface exhibited lower swelling degree and excellent stability. The hydrophilicity and antifouling performance of the modified membranes were improved dramatically. The water contact angle of M2-2 membrane decreased from the initial 36.6° ± 2.9 to 0° within 13 s. Static adsorption (14 h) of BSA for the M2-2 membrane was about 20.9 μg/cm^2^, decreasing by 76.2% compared to that of the pure membrane (88.0 μg/cm^2^). Moreover, for the long-term dynamic filtration experiments, the modified membranes showed excellent resistance to the three organic foulants (BSA, HA, and SA). Herein, M2-2 membrane exhibited the best antifouling properties: FRR of the above foulants maintained above 90% after one-cycle filtration (2 h). FRR of BSA solution was still as high as 90% even after 6 h cycling filtration, while *R*_t_ and *R*_ir_ were merely 10.7% and 3.3%, respectively. Therefore, the superior antifouling property and low-cost preparation of the hydrogel-modified PVDF membrane has a potential application in water treatment and protein purification.

## Figures and Tables

**Figure 1 membranes-11-00761-f001:**
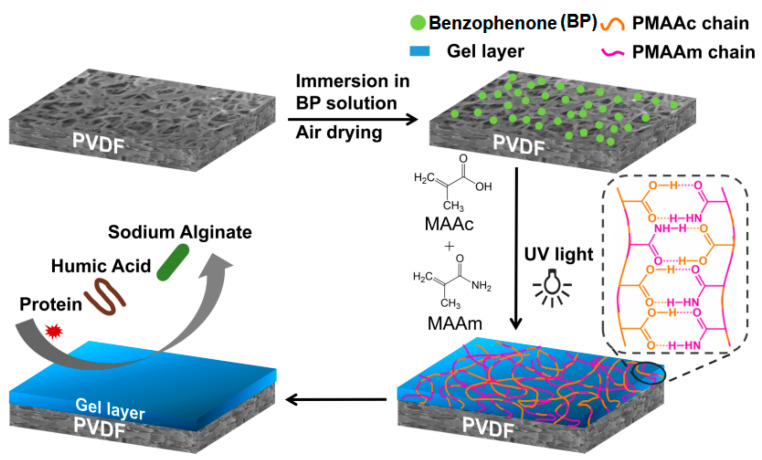
Schematic diagram for modification route of P(MAAc-*co*-MAAm)-grafted Poly(vinylidene fluoride) (PVDF) membrane.

**Figure 2 membranes-11-00761-f002:**
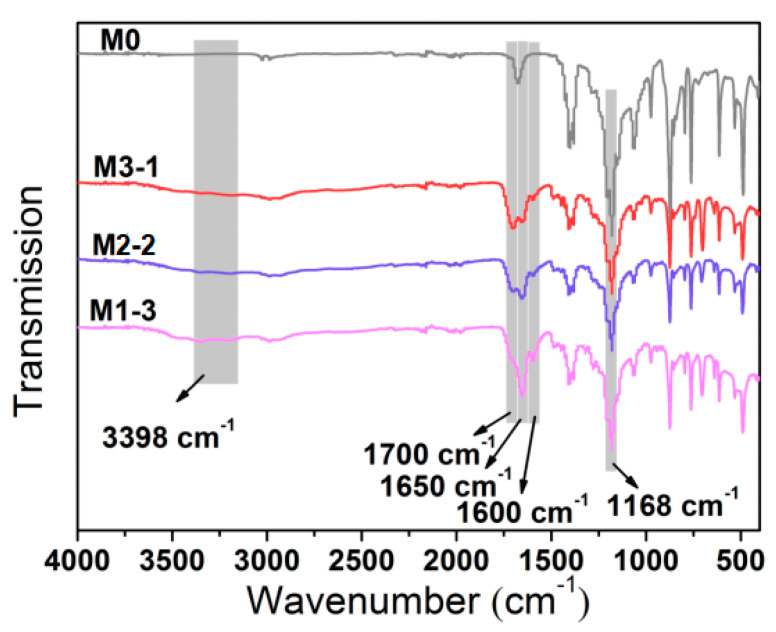
Attenuated total reflection Fourier transform infrared spectrum (ATR-FTIR) spectra of M0, M3-1, M2-2, and M1-3.

**Figure 3 membranes-11-00761-f003:**
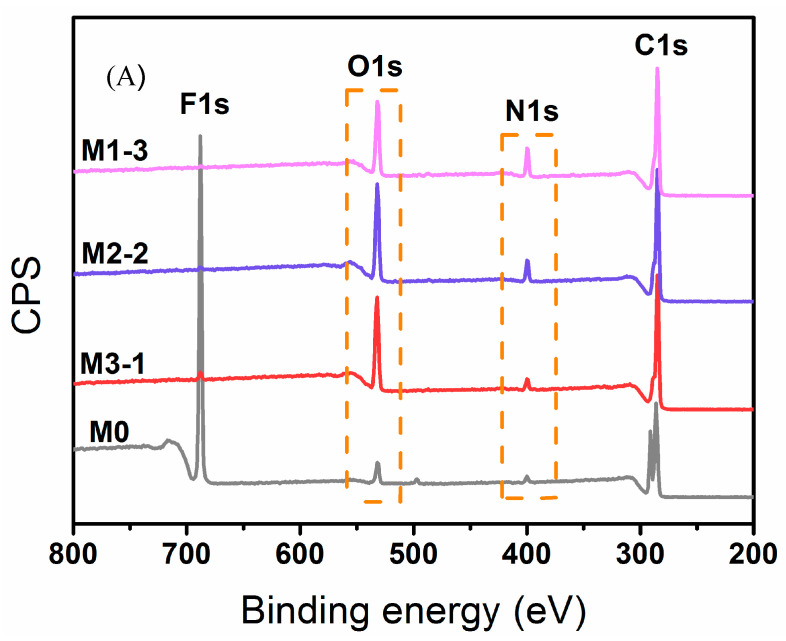
(**A**) X-ray photoelectron spectroscopy (XPS) spectra of membranes and (**B**) C 1 s core-level XPS spectra of membranes.

**Figure 4 membranes-11-00761-f004:**
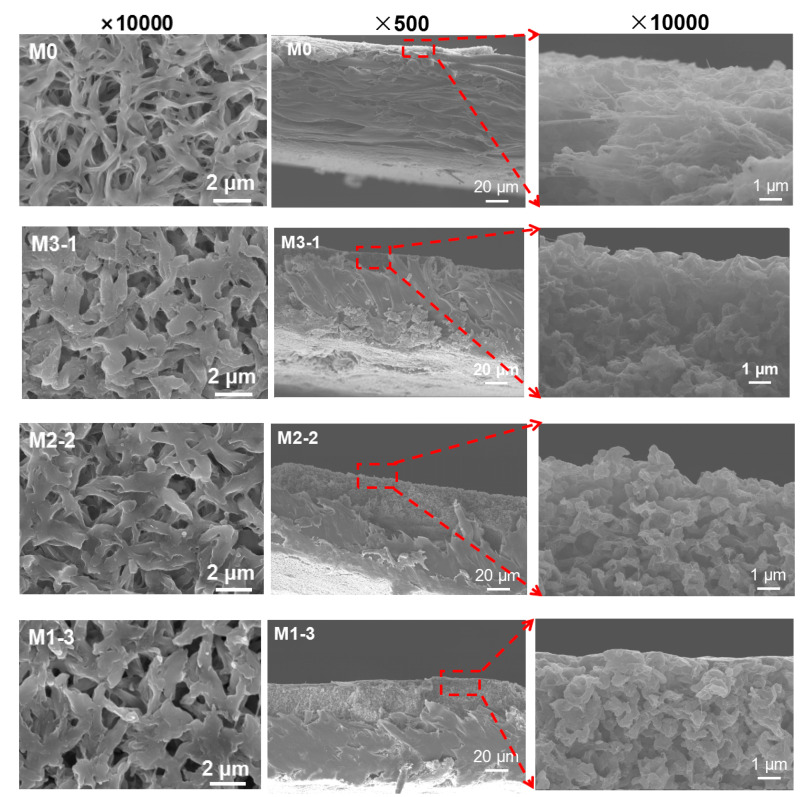
SEM images of membrane surface (**left**, ×10,000) and cross section (**middle**, ×500, and **right**, ×10,000) for M0, M3-1, M2-2, M1-3.

**Figure 5 membranes-11-00761-f005:**
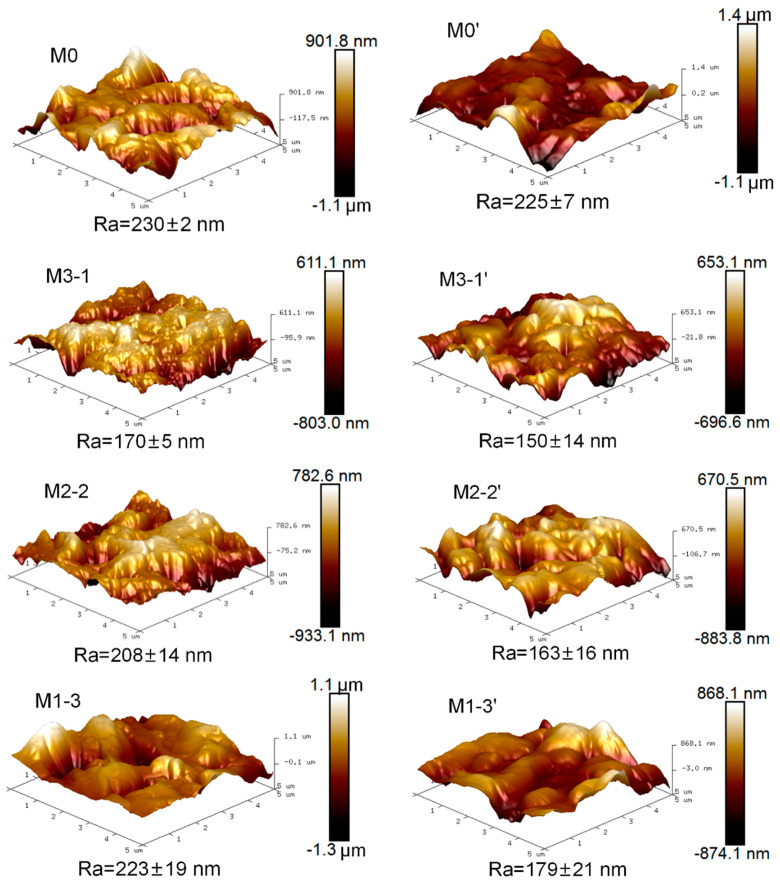
AFM photographs of membrane surfaces: (**left**) dry and (**right**) humid.

**Figure 6 membranes-11-00761-f006:**
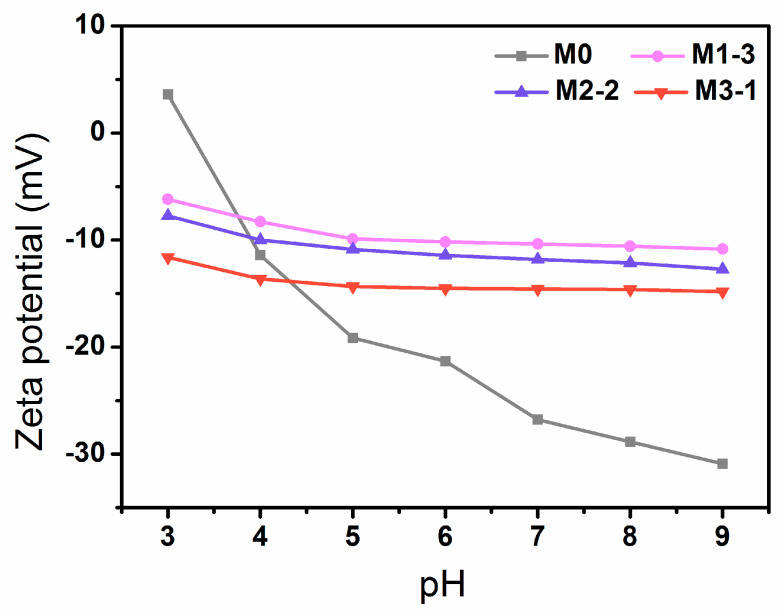
Zeta potential of original membrane and modified membranes.

**Figure 7 membranes-11-00761-f007:**
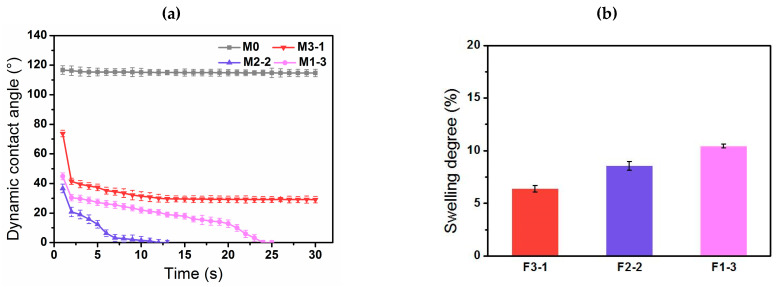
(**a**) Time-dependent water contact angle of membrane surfaces and (**b**) swelling degree of hydrogel layers on films (without pores).

**Figure 8 membranes-11-00761-f008:**
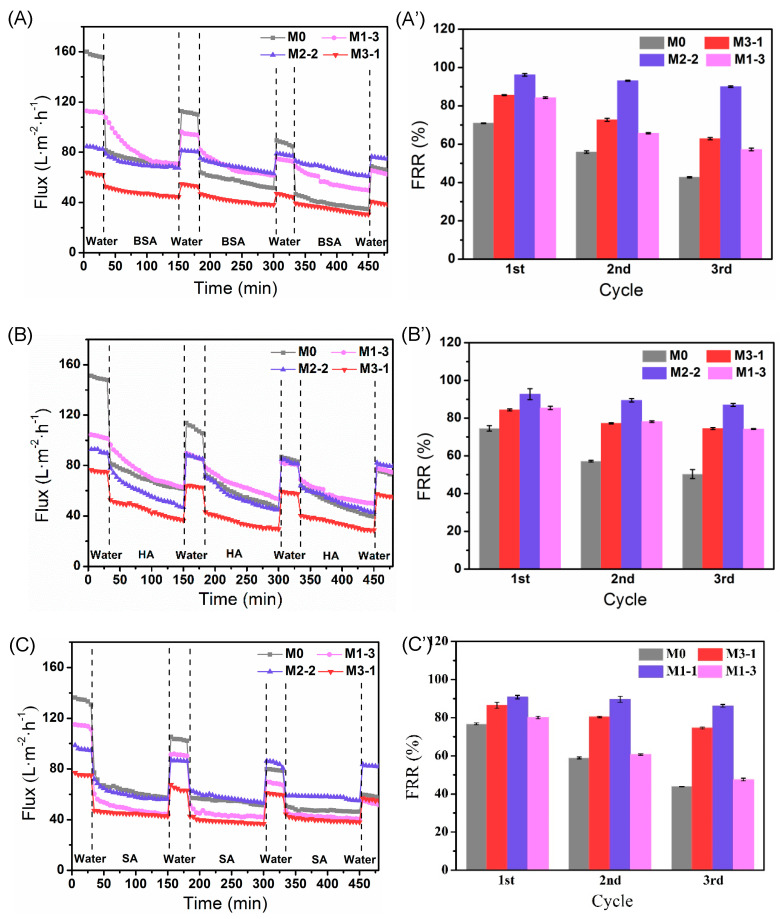
Time-dependent flux of three recycles of membranes for (**A**) BSA, (**B**) HA, and (**C**) SA; and the corresponding flux recovery ratio (FRR) for (**A’**) BSA, (**B’**) HA, and (**C’**) SA, respectively.

**Figure 9 membranes-11-00761-f009:**
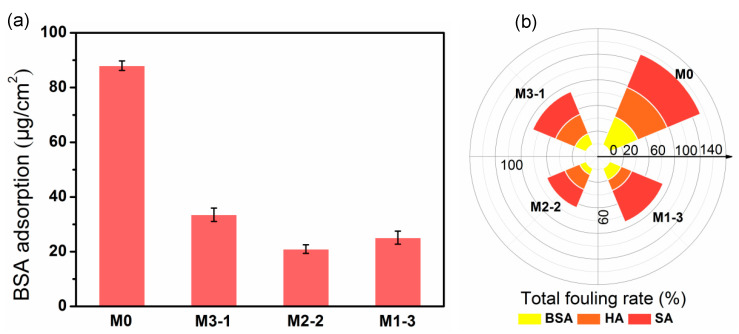
(**a**) Amounts of protein adsorbed onto membranes and (**b**) total fouling rate (*R*_t_) of dynamic filtration.

**Table 1 membranes-11-00761-t001:** Surface elemental composition of membranes.

Sample	Relative Element Content (at%)			
	C	N	O	F
M0	57.54	1.79	3.92	36.75
M3-1	73.26	4.08	21.15	1.51
M2-2	72.57	6.56	20.09	0.78
M1-3	73.51	8.70	17.58	0.21

**Table 2 membranes-11-00761-t002:** Comparison of antifouling ability of hydrophilic modified membranes in literature and M2-2 membrane in this work for three foulants.

Foulant	Hydrogel Material	Support	Preparation Method	One Cycle Filtration (min)	FRR (%)	*R*_t_ (%)	*R*_ir_ (%)	CA (°)	Ref.
BSA	P(MAAc-*co*-MAAm)	PVDF	UV-initiated radical graft polymerization	120	96.7	10.7	3.3	36.6	this work
	TiO_2_, PVA	PVDF	chemically binding	120	86.4	48.0	13.6	24.0	[5]
	VSA, METMAC	PES	UV photoinitiation	60	85.0	30.0	15.0	62.0	[21]
	APT	PVDF	blending	60	78.9	-	21.1	63.0	[42]
	HEA	PVDF	γ ray radiation	2	75.0	-	25.0	59.3	[42]
	CBMA	PVDF	physisorbed free radical polymerization grafting	60	92.0	44.0	8.0	70.2	[29]
	PEGMA	PVDF	radical grafting	60	90.8	55.5	9.2	60.0	[37]
HA	P(MAAc-*co*-MAAm)	PVDF	UV-initiated radical graft polymerization	120	92.8	24.9	7.2	36.6	**this work**
	TMC, SA	PVDF	crosslinking	60	91.0	35.0	8.7	36.0	[39]
	sulfonated polyaniline	PVDF	grafting	60	95.0	65.0	5.3	29.0	[42]
	PANI, MWCNT	PVDF	blending	120	85.0	79.0	15.0	54.8	[42]
	PEG, PD	PVDF	coating	60	89.1	-	10.9	61.5	[42]
	PVA, Glutaraldehyde	PES	coating	150	89.0	-	11.0	24.0	[42]
SA	P(MAAc-*co*-MAAm)	PVDF	UV-initiated radical graft polymerization	120	90.8	30.8	9.2	36.6	**this work**
	VSA, METMAC	PES	UV photoinitiation	60	86.0	-	14.0	62.0	[21]
	HEA	PVDF	γ ray radiation	2	92.0	-	11.5	59.3	[43]
	DMAPS	PVDF	UV-initiated radical graft polymerization	30	41.8	88.6	58.2	28.0	[22]

## Data Availability

The data presented in this study are available on request from the corresponding author.

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
