# Peer review of "Facile Fabrication of Multi-Hydrogen Bond Self-Assembly Poly(MAAc-co-MAAm) Hydrogel Modified PVDF Ultrafiltration Membrane to Enhance Anti-Fouling Property"

_membranes, 2021, doi:10.3390/membranes11100761_

Round 1

Reviewer 1 Report

The Manuscript (membranes- 1381046) entitled "Facile fabrication of multi-hydrogen bond self-assembly poly(MAAc-co-MAAm) hydrogel modified PVDF ultrafiltration membrane to enhance anti-fouling property" submitted to membranes focused on the improvement of antifouling of a commercial PVDF membrane. The authors tried to modify the surface membrane by UV grafting method with methacrylic acid (MAAc) and methyl acrylamide (MAAm) as hydrophilic monomers, and followed by multi- hydrogen bond self-assembly. The obtained results are interesting but the minor revision is needed according to the following comments prior to publishing.

# The supporting information was not received. Thus, it is not reviewed.

# In Table-1, the theoretical ratio of O/C, N/C and F/C of M0, M3-1, M2-2 and M1-3 should be added and then compared and discussed by the experimental values.

# The Ra of dry membranes are higher than the wet membranes but the reasons are not properly discussed.

# The correlation between swelling degree and water contact angle results was not discussed. Also, what is the effect of swelling degree on fouling resistance? It should be explained in the manuscript.

# The absolute value of Zeta potential of pristine membrane is very high. So, it is expected that PVDF membrane presents a good antifouling properties due to the high negative surface potential. In addition, it is needed the zeta potential of original membrane (PVDF membrane) compared with the reliable references in the literature.

# Why the dynamic filtration (cyclic filtration experiment) was not performed for a long time (at least 24 hours) to show the stability of membranes?

Reviewer 2 Report

The manuscript title “Facile fabrication of multi-hydrogen bond self-assembly poly(MAAc-co-MAAm) hydrogel modified PVDF ultrafiltration membrane to enhance anti-fouling property” reported by Fu et al. the synthesis of PVDF modified membrane was using UV radiation. As-prepared membranes were investigated in physicochemical properties using a various analytical tools. Moreover, PVDF modified membrane was analyzed the BSA rejection of anti-fouling properties. I recommend the few comments below;

  1. Should be improve the induction selection.
  2. The authors rewrite the preparation of PVDF modification clearly.
  3. Should be check the table 1, at% value is not corrected in the present table 1.
  4. Could you check the zeta potential value once again because didn’t see the variable in the all pH of modified membrane.
  5. Figure 6b, its correct. The authors check the swelling degree (%).
  6. The authors are considering to clear inside label in Figure 7. The reader is more confusing in the label form.
  7. Figure 8, Didn’t see the HA and SA absorption data bar results.
  8. The authors should be explaining the novelty of this work.
  9. Should be improve the abstract part.
  10. The authors check the typo error and a few places grammatical errors in the overall manuscript, and journal formant reference checks once again.

Reviewer 3 Report

In this work, the authors explained the facile fabrication of multi-hydrogen bond assembly poly (MAAc-co-MAAm) hydrogel modified PVDF ultra filtration membrane to enhance anti-fouling property. This article addressed about self-assembly poly (MAAc-co-MAAm) hydrogel modified PVDF for ultra-filtration technology. Overall, this work is quite interesting, even though there are some minor issues should be addressed before published on “Membranes”

This article can be improved by addressing the following issues.

  1. The manuscript should improve the typo errors and some grammatical errors.
  2. In the introduction part, the authors should add some more recent achievements.
  3. Authors should provide changes in the reaction observation and reaction mechanism for preparation of P (MAAc-co-MAAm) grafted PVDF membranes.
  4. Authors should measure the water uptake studies and compare the membrane porosity.
  5. What is the thickness of the prepared P (MAAc-co-MAAm) grafted PVDF membranes?
  6. Authors can include the contact angle images and compare the membrane swelling degree.
  7. Authors should calculate the solute rejection performance ratio (%) and compare the fouling resistant performance.
  8. Authors should tabulate the values of hydraulic resistance (Rm), surface roughness (nm) for pure and prepared membranes.
  9. There is a closely related reference for application of modified PVDF that need to be cited in Introduction part; Doi.org/10.1016/j.memsci.2018.03.049.
